# Numerical Simulations of Space Charge Waves Amplification Using Negative Differential Conductance in Strained Si/SiGe at 4.2 K

**Abel Garcia-Barrientos** [1,*], **Natalia Nikolova** [2], **Lado Filipovic** [3], **Edmundo A. Gutierrez-D.** [4], **Victoria Serrano** [5], **Sharon Macias-Velasquez** [6] and **Sarai Zarate-Galvez** [7]

1 Faculty of Science, Universidad Autónoma de San Luis Potosí (UASLP), San Luis Potosi 78294, Mexico
2 Department of Electrical and Computer Engineering, McMaster University, Hamilton, ON L8S 4K1, Canada; talia@mcmaster.ca
3 CDL for Multi-Scale Process Modeling of Semiconductor Devices and Sensors, Institute for Microelectronics, TU Wien, 1040 Vienna, Austria; filipovic@iue.tuwien.ac.at
4 Electronics Department, INAOE, Puebla 72840, Mexico; edmundo@inaoep.mx
5 Grupo de Investigacion en Tecnologias Computacionales Emergentes, Chiriqui Regional Center, Universidad Tecnologica de Panama, Lassonde, David 040601, Chiriqui, Panama; victoria.serrano@utp.ac.pa
6 Faculty of Engineering, Universidad Autónoma de San Luis Potosí (UASLP), San Luis Potosi 78294, Mexico; sharon.macias@uaslp.mx
7 Instituto de Investigación en Comunicación Óptica (IICO), Universidad Autónoma de San Luis Potosí (UASLP), San Luis Potosi 78294, Mexico; a322674@alumnos.uaslp.mx
* Correspondence: abel.garcia@uaslp.mx

**Abstract:** This paper introduces a two-dimensional (2D) numerical simulation of the amplification of space charge waves using negative differential conductance in a typical MOS silicon–germanium (SiGe)-based field-effect transistors (FET) and complementary metal oxide semiconductor (CMOS) technology at 4.2 K. The hydrodynamic model of electron transport was applied to describe the amplification of space charge waves in this nonlinear medium (i.e., the negative differential conductance). This phenomenon shows up in GaAs thin films at room temperature. However, this can be also observed in a strained Si/SiGe heterostructure at very low temperatures (T < 77 K) and at high electric fields (E > 10 KV/cm). The results show the amplification and non-linear interaction of space charge waves in a strained Si/SiGe heterostructure occurs for frequencies up to approximately 60 GHz at T = 1.3 K, 47 GHz at T = 4.2 K, and 40 GHz at T = 77 K. The variation of concentration and electric field in the Z and Y directions are calculated at 4.2 K. The electric field in the Z direction is greater than in the Y direction. This is due to the fact that this is the direction of electron motion. In addition to deep space applications, these types of devices have potential uses in terrestrial applications which include magnetic levitation transportation systems, medical diagnostics, cryogenic instrumentation, and superconducting magnetic energy storage systems.

**Keywords:** space charge waves; SiGe; negative differential conductance



## 1. Introduction

In the past five decades, extensive research has been conducted on the amplification of space charge waves through the utilization of negative differential conductance (NDC) [1–5]. These devices find applications in a wide range of fields, such as amplifiers, oscillators, and frequency multipliers, particularly in the millimeter and sub-millimeter wave range. By harnessing the negative differential conductance phenomenon as a nonlinear medium, these devices can generate harmonics of input signals. The significance of these applications lies in their crucial role in high-frequency communication systems and photonics. Numerous materials, including GaAs, GaN, InP, and InN, exhibit the negative differential conductance phenomenon, when subjected to high electric fields at room temperature [6]. Unfortunately,

most of these compound materials are incompatible with silicon technology. However, a strained Si/SiGe heterostructure offers compatibility with MOSFET technology and it demonstrates negative differential conductance as well, albeit only at extremely low temperatures—specifically at 77 K, 4.2 K, and 1.3 K [7–9]. In a previous study by Garcia-Barrientos et al. [10], a numerical investigation using simulations of space charge wave propagation in strained Si/SiGe heterostructures at 77 K was conducted. Additionally, two-dimensional (2D) simulations of space charge wave amplification in a strained Si/SiGe heterostructure at 77 K were presented in a separate study [11]. However, these articles solely focused on numerical simulations of space charge wave amplification at 77 K without considering calculations for temperatures at 4.2 K and without performing a parametric interaction analysis.

The mentioned research works utilized electron mobility calculations at various temperatures as a function of the effective electric field, sourced from references [12,13]. Nevertheless, it is worth noting that the negative differential conductance exhibits a larger dynamic range at temperatures of 4.2 K and 1.3 K, as revealed in [14]. Therefore, our research work aims to present 2D numerical simulations of space charge wave amplification in strained Si/SiGe heterostructures at very low temperatures—specifically at 4.2 K—where the negative differential conductance manifests. To accomplish this, we employed the hydrodynamic model to analyze electron transport at high fields in strained Si/SiGe heterostructures [15]. By utilizing the finite difference time domain (FDTD) method, we solved the hydrodynamic model in conjunction with the Poisson equation. Through this approach, we obtained the output spectrum of the input signal and calculated the variation of the electric field in both the Z and Y directions. The electric field was found to be greater in the Z direction, as the electrons predominantly move along this axis. Our results indicate the potential for developing cryogenic CMOS-based mm-wave amplifiers and frequency multipliers, presenting exciting possibilities for new satellite communication systems.

Negative differential conductance (NDC) is a fascinating phenomenon observed in a wide range of electronic systems, wherein the current passing through the system decreases as the applied voltage is increased. This behavior defies the classical understanding of Ohmic conductance, where an increase in voltage typically leads to an increase in current. NDC represents a unique and intriguing characteristic, particularly notable in semiconductor devices. Essentially, NDC refers to a two-terminal component capable of amplification, converting DC power applied to its terminals into AC output power to amplify an AC signal applied to the same terminals. The discovery of NDC has paved the way for the development of novel electronic devices, including oscillators and amplifiers, with a particular focus on microwave frequencies. These devices play a critical role in communication, sensing, and computing systems by offering enhanced efficiency, speed, and sensitivity. The underlying physics behind NDC can be attributed to phenomena such as inter-valley transfer at high electric fields, avalanching breakdown, and the quantum mechanical properties of the electronic system. In essence, NDC arises from the interaction between electronic states within the system and the external voltage bias. The presence of resonant states is closely linked to the occurrence of NDC. Resonant states manifest as sharp peaks in the density of states within the system. These resonant states can lead to the formation of a current bottleneck, resulting in a reduction of current flow as the applied voltage increases. In recent years, significant progress has been made in comprehending and controlling NDC behavior across various electronic systems [1]. Further research in this field is anticipated to yield the development of new materials and devices with improved performance and functionality, especially within the realm of strained Si/SiGe heterostructures [16–18].

In this study, we focus on utilizing the strained Si/SiGe heterostructure (Figure 1), which is a semiconductor structure composed of alternating layers of silicon and silicon-germanium (SiGe) films. These layers possess distinct properties, including different crystal structures and electronic band structures, resulting in intriguing electrical characteristics at their heterojunctions. One remarkable attribute of strained Si/SiGe heterostructures is their

ability to exhibit negative differential conductance [7]. This phenomenon strongly depends on the thickness of the SiGe layer and the applied voltage and can be finely adjusted by modifying the composition and thickness of the SiGe layer. The tunability of negative differential conductance in strained Si/SiGe heterostructures presents an opportunity to design custom devices with specific performance attributes. By precisely controlling the composition and thickness of the SiGe layer, the NDC behavior can be tailored to meet specific design requirements. Consequently, strained Si/SiGe heterostructures have emerged as a versatile semiconductor platform, facilitating band structure and strain engineering. This, in turn, enables the enhancement of conventional microelectronic device performance and the exploration of novel concepts. In essence, strained Si/SiGe heterostructures offer a unique semiconductor structure capable of exhibiting negative differential conductance. This property has opened avenues for band structure and strain engineering, enabling divers.

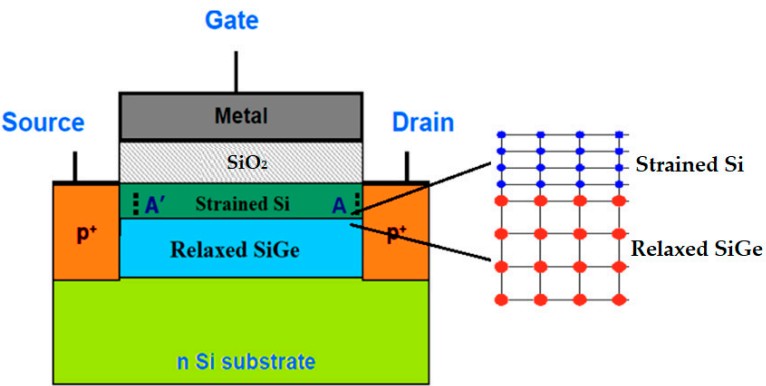

**Figure 1.** Schematic structure of a strained Si/SiGE MOSFET. A′ and A are the input and output coupling element, respectively.

In summary, a 2D numerical simulation of the amplification of space charge waves using negative differential conductance in a typical MOS silicon–germanium (SiGe)-based field effect transistors (FET) and complementary metal oxide semiconductor (CMOS) technology at 4.2 K is presented in this paper. The results show the amplification and non-linear interaction of space charge waves in a strained Si/SiGe heterostructure occurs for frequencies up to approximately 60 GHz at T = 1.3 K, 47 GHz at T = 4.2 K, and 40 GHz at T = 77 K. In addition to deep space applications, these types of devices have potential uses in terrestrial applications which include magnetic levitation transportation systems, medical diagnostics, cryogenic instrumentation, and superconducting magnetic energy storage systems. As a future work, the 2D analysis for amplification of space charge waves in strained Si/SiGe heterostructure at 1.3 K will be interesting.

## 2. Device and Method

The strained Si/SiGe MOSFET shown in Figure 1 demonstrates the propagation and amplification of space charge waves on the surface of the Si/SiGe inversion (100) layer, when the gate voltage surpasses the inversion threshold. To induce the phenomenon of negative differential conductance, a bias electric field of $E_0 > 10\,\mathrm{kVcm^{-1}}$ is applied between the source and drain, where $dv/dE < 0$. At millimeter and submillimeter wavelengths, the input coupling element (input A′) excites space charge waves, which are subsequently amplified and collected at the output coupling element (output A) on the strained Si/SiGe layer. There, a thin layer of Si is grown on a relaxed $Si_{1-x}Ge_x$ buffer, the Si layer is forced to assume the larger lattice constant of the underlying $Si_{1-x}Ge_x$ substrate, see Figure 1. The Si layer is thus said to be under biaxial tensile strain where the amount of strain is given by the Ge content (denoted as x) in the substrate [19]. High quality strained Si/SiGe heterostructures can be fabricated by molecular beam epitaxy or by chemical vapor deposition [20–23].

In this system, high-field electron transport within the strained Si/SiGe MOS inversion layer is primarily governed by intervalley phonon scattering involving the sixfold degenerate valleys. To model this electron transport, we employ the quasi-hydrodynamic model in conjunction with Poisson's equation. The strain-induced band ordering in the Si/SiGe structure allows for the occupation of only two valleys, with an effective transport mass $m_t$ = 0.19 $m_0$ along the channel and a quantization mass $m_l$ = 0.92 $m_0$ perpendicular to the channel. Consequently, we can describe high-field transport using a two-valley model.

Under these conditions, electrons can acquire sufficient energy from the electric field to transition from the lower to the upper valley. We define $n_1$ and $n_2$ as the respective electron counts in bands 1 (lower) and 2 (upper). Thus, two sets of balance equations are required to model this system accurately. The conservation of particles in this uniform two-valley system is expressed through the following continuity equations, see Equations (1) and (2).

$$\frac{\partial n_1}{\partial t} = -\frac{n_1}{\tau_{12}} + \frac{n_2}{\tau_{21}} \tag{1}$$

$$\frac{\partial n_2}{\partial t} = -\frac{n_2}{\tau_{21}} + \frac{n_1}{\tau_{12}} \tag{2}$$

The scattering time from valley $i$ to $j$ is denoted as $\tau_{ij}$. For our analysis, we assume that electrons in valleys 1 and 2 are in equilibrium, meaning they have the same electron temperature. Although this assumption is not entirely accurate, it greatly simplifies the analysis process. In the two-valley system, momentum conservation is maintained through the following equations, see Equations (3) and (4),

$$\frac{dp_1}{dt} = qE - \frac{p_1}{\tau_{m1}(T_e)} - \frac{p_1}{\tau_{12}(T_e)} \tag{3}$$

$$\frac{dp_2}{dt} = qE - \frac{p_2}{\tau_{m2}(T_e)} - \frac{p_2}{\tau_{21}(T_e)} \tag{4}$$

where $p_i = m_i v_i$ is the momentum and $\tau_{mi}$ is the momentum scattering rate in the $i$ valley. Since we are assuming that valleys 1 and 2 are at equilibrium, we can rely on the simple one-valley energy conservation, see Equation (5)

$$\frac{\partial}{\partial t}\left(n\frac{3}{2}k_B T_e\right) = nqv_d E - n\frac{3}{2}k_B \frac{T_e - T_0}{\tau_E}, \tag{5}$$

where $T_e$ is the electron temperature, $T_0$ is the lattice temperature, $\tau_E$ is the energy relaxation time, with $n = n_1 + n_2$ and $J = qnv_d = q(n_1 v_1 + n_2 v_2)$ [18]. For our simulations, we need calculations of drift velocity and the average electron energy as functions of the electric field. The two-valley model uses the single-electron–gas approximation and accounts for all intervalley transfer effects through the effective value for carrier velocity, energy, effective mass, and relaxation rates. We have chosen 0.25 eV as the energy separation between the lower and upper valleys. Depending on the strain, this energy separation factor may yield different values [23].

Figure 2a presents the computation of the steady-state drift velocity as a function of electric field at temperatures of 77 K, 4.2 K [9], and 1.3 K [7]. Beyond electric fields of 10 kV/cm, negative differential conductivity is observed at these temperatures. This behavior arises due to the transition of electrons from the twofold valley to the fourfold valley, where the effective mass is larger. At 300 K, the drift velocity reaches a notable saturation point of $1 \times 10^7$ cm/s at 10 kV/cm due to the low effective mass of electrons in the twofold valleys and reduced intervalley scattering of phonons between the twofold and fourfold valleys [24,25]. Consequently, negative differential conductivity is absent. The presence of negative differential conductivity at temperatures of 77 K, 4.2 K, and 1.3 K ensures the amplification of space charge waves, similar to the behavior observed

in n-GaAs thin films [1]. In Figure 2b, the average electron energy at 300 K, 77 K, and 4.2 K is plotted as a function of the electric field. The energy is measured relative to the bottom of the lowered valleys (valley pair 1 in the current model), and the sum of kinetic energy and potential energy (0 in the lower valley and ΔE in the upper valleys) is averaged and plotted. At thermal equilibrium, the energy is $3kT/2$ = 39 meV at 300 K and increases monotonically with the field. The energy exhibits a clear dependence on ΔE, although this dependency diminishes at fields larger than 100 kV/cm. In higher fields, electrons possess an energy greater than ΔE, resulting in minimal differences for different ΔEs. It should be noted that as ΔE increases, impact ionization becomes more significant, as the band gap and threshold decrease with ΔE. Figure 2b illustrates the energy-field characteristics at 77 K, comparing it with the same plot at 300 K. The electron energy is lower up to a field of 10 kV/cm, after which the order is reversed in higher fields, contributing to higher mobility at 77 K. However, an important consideration influencing performance is the temperature-dependent electron transport. Specifically, at ultra-low temperatures around 1.3 K, understanding the behavior of electron transport in strained Si/SiGe heterostructures becomes particularly important. At these temperatures, electron transport is primarily limited by tunneling through the SiGe barrier. However, at extremely low temperatures, electrons become strongly localized, resulting in a decrease in tunneling current. This phenomenon, known as Coulomb blockade, occurs due to the charging energy of electrons trapped in quantum dots, leading to increased resistance. Comprehending temperature-dependent electron transport in SiGe heterostructures at ultra-low temperatures is vital for designing and optimizing devices operating in this regime. Ongoing research in this area is expected to provide further insights into the underlying physics of electron transport in SiGe heterostructures and enhance the performance of future electronic devices. However, analyzing space charge waves in strained Si/SiGe heterostructures at 1.3 K remains an open topic, as additional important physical phenomena must be considered.

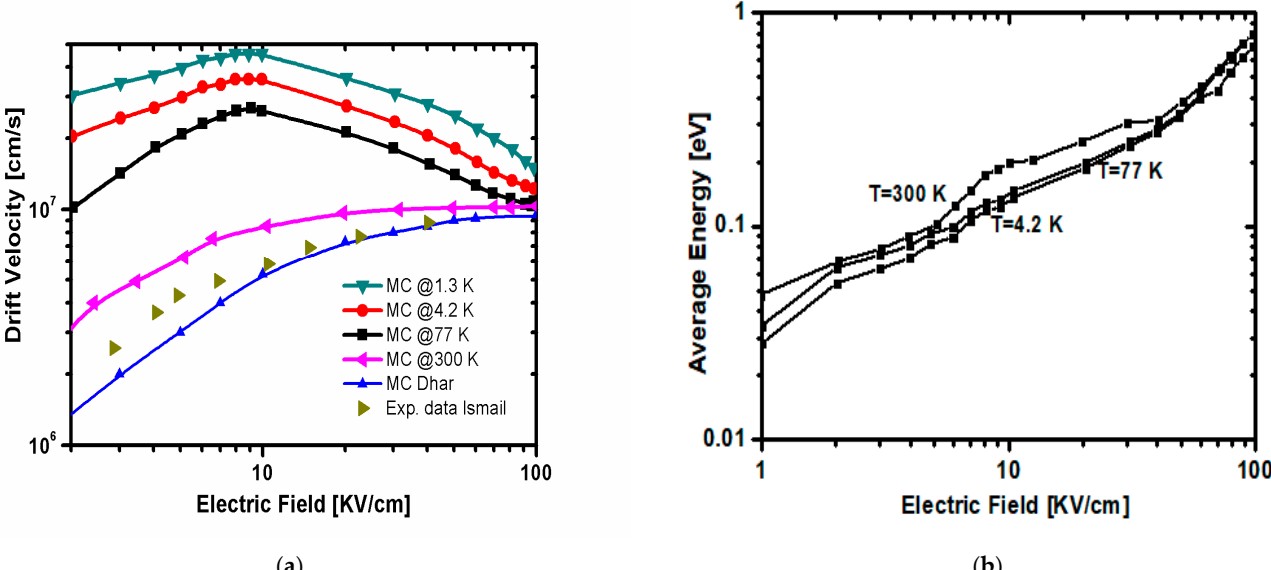

(**a**)  (**b**)

**Figure 2.** The steady-state drift velocity as a function of electric field at temperatures of 300 K, 77 K, 4.2 K and 1.3 K (**a**). Energy-field as a function of electric field at temperatures of 300 K, 77 K, 4.2 K (**b**).

The drift velocity at 77 K is constrained by phonon scattering, resulting in a decrease in electron mobility. As the temperature lowers to 4.2 K, the impact of phonon scattering diminishes, leading to an increase in the drift velocity. At ultra-low temperatures of 1.3 K, electron transport is predominantly limited by tunneling through the material, resulting in a substantial rise in the drift velocity with increasing electric field. The temperature-dependent variation of drift velocity plays a critical role in the design and optimization of electronic devices. Understanding this relationship enables the prediction of electron

behavior in different temperature conditions and facilitates the design of devices tailored to specific temperature regimes.

## 3. Results

We explore the behavior of space charge waves in a strained Si/SiGe heterostructure at a temperature of 4.2 K, focusing on the range of frequencies and wave propagation characteristics. By applying an electric field bias parallel to the surface, in the direction of propagation, we achieve negative differential conductivity with $E_0 > 10.3$ kV/cm. Our analysis involves studying the dispersion equation $D(\omega,k) = 0$, which establishes a relationship between the frequency ($\omega$) and longitudinal wave number ($k$). We consider cases where $\omega$ is real and $k$ takes the form of $k' + ik''$. A positive imaginary part of $k''$ indicates spatial increment or amplification, while a negative value signifies decrement or damping. Our findings, illustrated in Figure 3, demonstrate that amplification of space charge waves occurs for frequencies up to approximately 60 GHz at T = 1.3 K, 47 GHz at T = 4.2 K, and 40 GHz at T = 77 K. For more comprehensive insights at 77 K, refer to Ref. [11]. Notably, at 4.2 K, $k''$ becomes negative for frequencies above 47 GHz, indicating the possibility of space charge wave propagation without amplification. This observation suggests that a strained Si/SiGe heterostructure has the potential for amplifying space charge waves within a frequency range similar to that of an n-GaAs thin film, albeit at low temperatures.

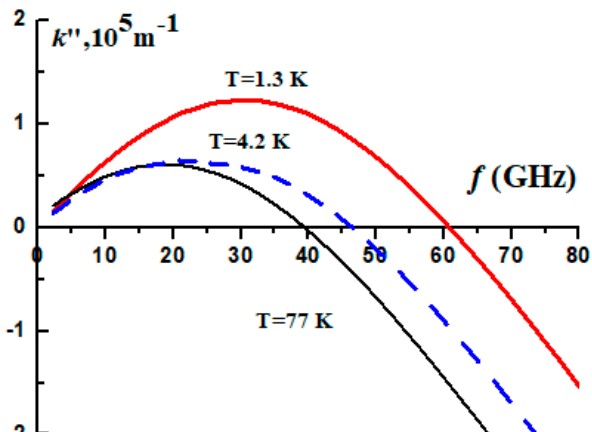

**Figure 3.** Spatial increment $k''(\omega)$ for tow-valley model for a strained Si/SiGe heterostructure at 77 K (blank), 4.2 K (blue) and 1.3 K (red).

To carry out the simulations, we introduce an input signal, and it was applied in the input coupling element, see Equation (6).

$$\widetilde{E}_{ext} = E_o sin(\omega t)exp\left(-\left(\frac{z-z_1}{z_0}\right)^2 - \left(\frac{y-y_1}{y_0}\right)^2\right) \tag{6}$$

The output signal clearly demonstrates the amplification effect. In this scenario, a microwave input signal with a frequency of 10 GHz is applied to the input coupling element, resulting in the excitation of space charge waves in the 2D electron gas. These waves undergo amplification due to the presence of negative differential conductivity. We employ a stable implicit difference scheme for our analysis, utilizing the parameters provided in Table 1.

**Table 1.** Parameters used in the simulations.

| Parameter | Symbol | Value |
|---|---|---|
| electron density | $n_0$ | $5 \times 10^{14}$ cm$^{-2}$ |
| drift velocity [1] | $v_0$ | $2 \times 10^7$ cm/s |
| film length | $L_z$ | 100 μm |
| film width | $L_y$ | 100 μm |
| film thickness | 2 h | 0.001–0.01 μm |
| drift velocity | $v_d(E)$ | |
| effective mass | $m(E)$ | |
| average energy | $w(E)$ | |
| frequency input | $f = \omega/2\pi$ | 1 to 60 GHz |

[1] corresponding to $E_0$ = 15–30 kV/cm.

Figure 4 illustrates the typical output spectrum of the electromagnetic signal. The input carrier frequency is set at $f$ = 15 GHz, with an input electric microwave signal amplitude of $E_m$ = 25 V/cm. Despite a decreasing growth rate with increasing *rf* frequency, our results show an amplification of 23 dB in this particular case. The input pulse duration is $2t_0$ = 2.5 ns, with the maximum of the pulse occurring at $t_1$ = 2.5 ns. The amplified signal at the first harmonic of the input signal is evident, as well as the generation of harmonics resulting from the non-linear behavior of space charge waves. Specifically, we observe the second and third harmonics with frequencies of 20 GHz and 30 GHz, respectively. While it may be possible to obtain the fourth and fifth harmonics, these cannot be effectively amplified, as shown in Figure 3, and they are perceived as noise. In Figure 4b, we explore parametric amplification by introducing two input signals with frequencies $f_1$ = 15 GHz and $f_2$ = 20 GHz, respectively. The resulting output signals are displayed. Here, we can observe the output signals, including the second harmonics, as well as the combined effect of the two input signals.

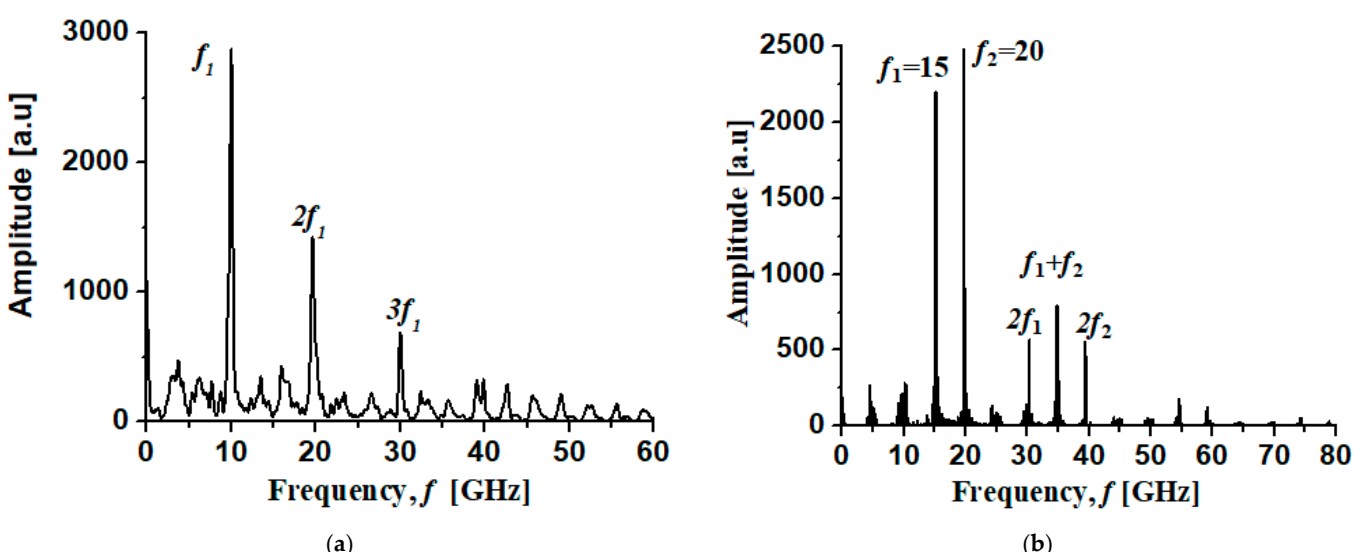

**Figure 4.** Output spectrum for one input signal with $f_1$ = 10 GHz (**a**). Output spectrum for two input signals with $f_1$ = 15 GHz and $f_2$ = 20 GHz (**b**).

Figure 5 depicts the spatial distributions of the alternate components of the electric fields ($\tilde{E}_z$ and $\tilde{E}_y$) and electron concentration $\tilde{n}$ at a specific time moment, $t$ = 4 ns. Notably, the maximum variations occur in the output coupling element, and the spatial distribution in the $\tilde{E}_z$ direction is larger than in the $\tilde{E}_y$ direction. This difference arises due to the propagation of the space charge wave. The transverse width of the film along the Y axis is 100 μm. These spatial distributions are presented for a time moment of 1.5 ns after the input signal reaches its maximum value, and the input electric pulse has a duration of 2.5 ns.

Through direct 2D numerical simulations, we have confirmed the linear amplification of space charge waves. Moreover, the possibility of non-linear frequency doubling and mixing has been demonstrated. For effective frequency doubling in the millimeter wave range, it is advisable to employ films with uniform doping. It is worth noting that the output signal also exhibits a frequency of 15 GHz.

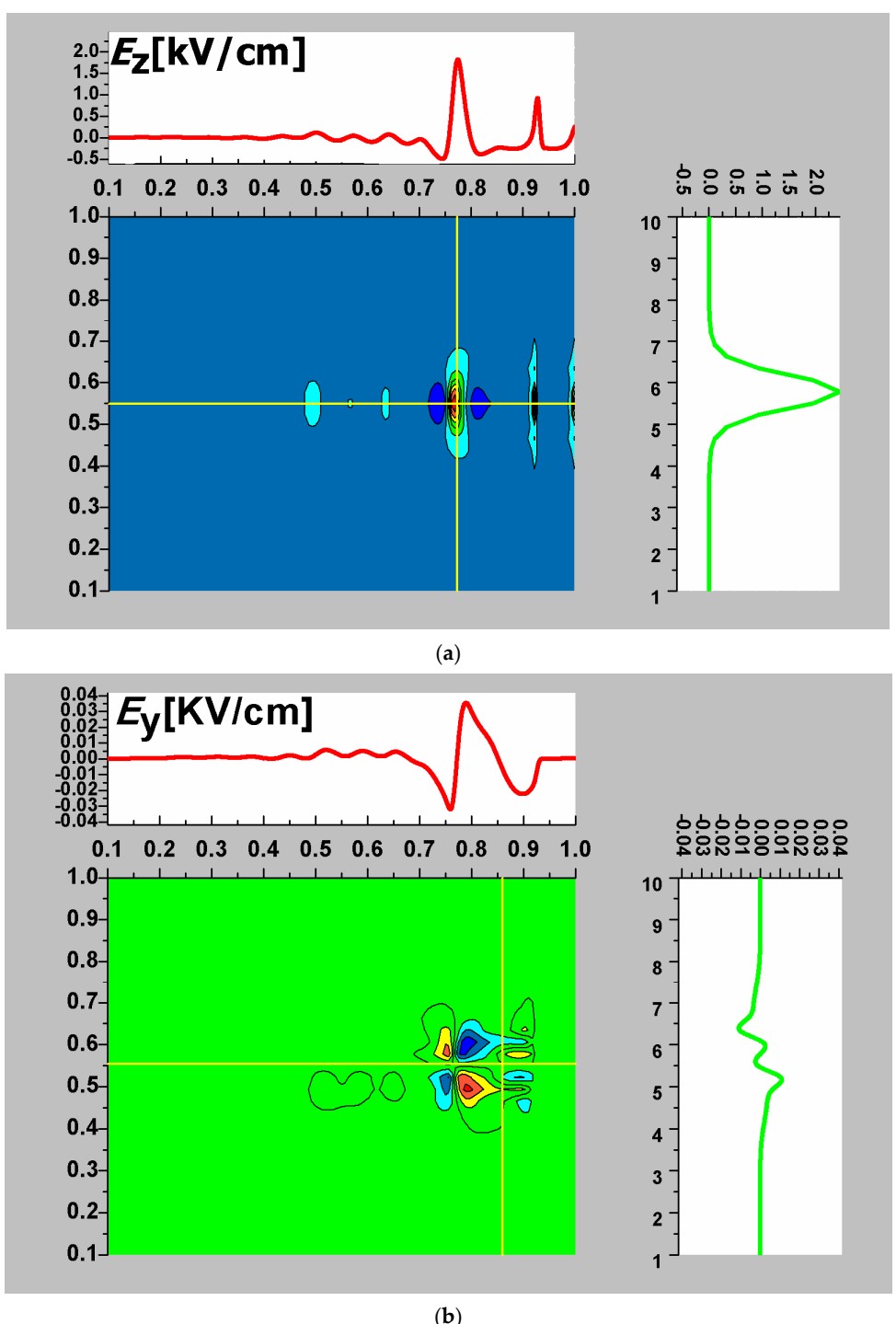

**Figure 5.** *Cont.*

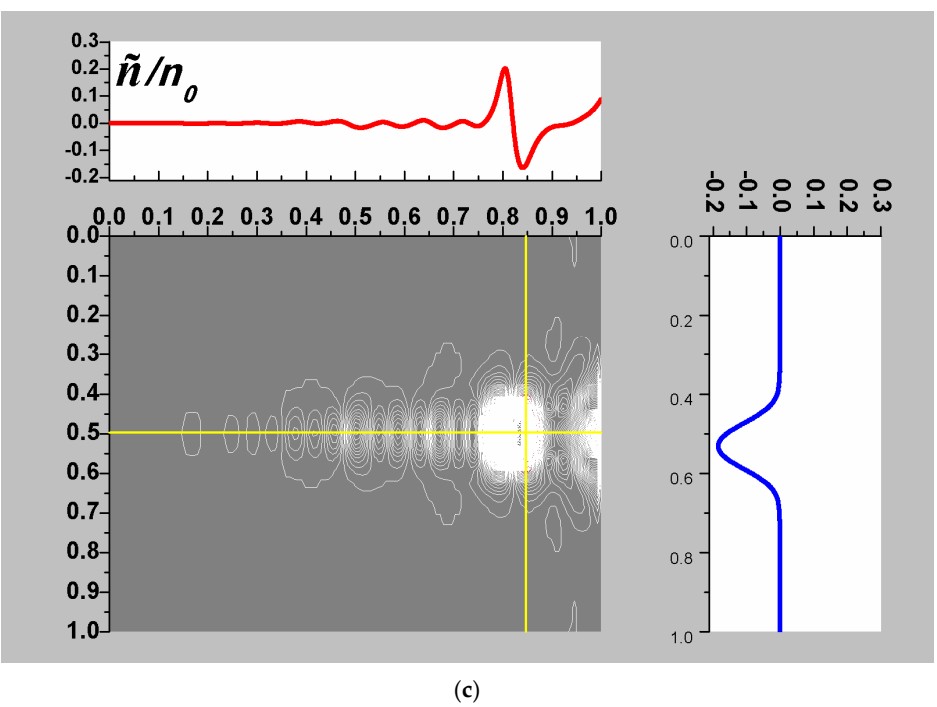

(**c**)

**Figure 5.** The spatial distributions of the alternative part of the electric field component $E\tilde{}_z$ of the apace charge wave at 4.2 K (**a**); the component of electric field $E\tilde{}_y$ at 4.2 K (**b**); alternative part of the electron concentration $\tilde{n}$ at 4.2 K (**c**).

## 4. Conclusions

This work presents a 2D numerical simulation aimed at studying the amplification of space charge waves utilizing negative differential conductance in a strained Si/SiGe heterostructure at a temperature of 4.2 K. The quasi-hydrodynamic model of electron transport is employed to investigate the amplification of space charge waves in this nonlinear medium, particularly focusing on the phenomenon of negative differential conductance. While negative differential conductance is commonly observed in GaAs thin films at room temperature, this study demonstrates its occurrence in strained Si/SiGe heterostructures even at very low temperatures. The results show that amplification and non-linear interaction of space charge waves in strained Si/SiGe heterostructure occur for frequencies up to approximately 60 GHz at T = 1.3 K, 47 GHz at T = 4.2 K, and 40 GHz at T = 77 K. Furthermore, the analysis of the electric field variation in the Z and Y directions indicates that the electric field along the Z direction surpasses that along the Y direction due to the movement of electrons in that specific direction, $E\tilde{}_z$ is nearly 50 times greater than $E\tilde{}_y$. It is worth noting that, although negative differential conductance can also be observed in strained Si/SiGe heterostructures at 1.3 K, it is crucial to acknowledge that the electron transport models differ significantly in this scenario. This emphasizes the need for further investigation into the underlying physical phenomena.

Overall, these findings provide valuable insights into the amplification of space charge waves and the role of negative differential conductance in strained Si/SiGe heterostructures, emphasizing their potential applications in low-temperature electronic devices and expanding our understanding of electron transport in these systems. Further investigations and refinements of the electron transport models are necessary to uncover the underlying mechanisms and enhance our knowledge of this intriguing physical phenomenon.

**Author Contributions:** Conceptualization, A.G.-B. and E.A.G.-D.; methodology, L.F.; software, N.N.; validation, S.M.-V., V.S. and S.Z.-G.; formal analysis, A.G.-B.; investigation, E.A.G.-D.; resources, N.N.; data curation, A.G.-B.; writing—original draft preparation, A.G.-B.; writing—review and editing, S.Z.-G.; visualization, N.N.; supervision, E.A.G.-D.; project administration, V.S.; funding acquisition, L.F. All authors have read and agreed to the published version of the manuscript.

**Funding:** This work was supported in part by CONACyT Mexico by Sabbatical fellowship at McMaster University and the PhD Scholarship with CVU number: 626570. Also, this work was supported by OEaD Agency at the Vienna University of Technology fellowship. Lado Filipovic gratefully acknowledges financial support through the Austrian Science Fund (FWF) grant P33609-N and the Austrian Federal Ministry of Labour and Economy, the National Foundation for Research, Technology and Development and the Christian Doppler Research Association.

**Conflicts of Interest:** The authors declare no conflict of interest.

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
