# Peer review of "Numerical Simulations of Space Charge Waves Amplification Using Negative Differential Conductance in Strained Si/SiGe at 4.2 K"

_crystals, doi:10.3390/cryst13091398_

Round 1
Reviewer 1 Report
Comment to the Author
The presented work entitled "Numerical Simulations of Space Charge Waves Amplification Using Negative Differential Conductance in Strained Si/SiGe at 4.2 K" discusses the 2D numerical simulation of the amplification of space charge waves using negative differential conductance in a strained Si/SiGe heterostructure at 4.2 K. The hydrodynamic model of electron transport was applied to describe the amplification of space charge waves in this nonlinear medium, i.e., the negative differential conductance. This phenomenon shows up in GaAs thin films at room temperature; however, this can be also observed in strained Si/SiGe heterostructure at very low temperatures. The results show the amplification and no-linear interaction of space charge waves in strained Si/SiGe heterostructure at 4.2 K. The manuscript may accept after incorporation of the following concerns but the current form is not actable.
1. The abstract should be rewritten as it should indicate the main findings (with some values).
2. Abstract should contain future application of work.
3. Introduction part should have more recent references.
4. At the last paragraph of introduction part, work should summarize.
5. Equations are not cited properly. Give the refence for each equation.
6. Process diagram for the fabrication of the device is required.
7. What parameters have been considered in this work. Show in a table.
8. In some of the sentence, it is difficult to understand the meaning of text.
9. Result explanation is not sufficient and justified. Give proper reason for all the results and justify.
10. Conclusion should have some resulting details with numerical data.
In some of the sentences, it is difficult to understand the meaning of the text.
Author Response
Dear Reviewer,
Thank you very much for your comments. They were excellent, and they help us to present the paper in the best form. Please, if you have any more comments, tell us as soon as possible.
Best regards

Reviewer 2 Report
Crystals-2596776
Review Report
The MS entitled “Numerical Simulations of Space Charge Waves Amplification Using Negative Differential Conductance in Strained Si/SiGe at 4.2 K” by Abel Garcia-Barrientos and co-workers presents a 2D numerical simulation of the amplification of space charge waves in a strained Si/SiGe heterostructure at 4.2 K.
Authors employ the hydrodynamic model to simulate the electron transport in strained Si/SiGe heterostructures. They used a finite difference time domain method to self-consistently solve the hydrodynamic model and the Poisson equation.
Authors must compulsory answer the following question before their paper could be considered for publication:
1 – Authors stated in the Abstract that “ … strained Si/SiGe heterostructure offers compatibility with heterostructure bipolar transistors”, but they develop all the work in the paper on n-channel FETs in which Si is strained on a relaxed SiGe buffer. Please, clarify this and also the statement on lines 91-92 “ … which is a semiconductor structure composed of alternating layers of silicon and silicon-germanium (SiGe) materials …”.; clearly in Figure 1 and across the paper no alternating structure is neither studied nor mentioned.
2 – Please, for the sake of clarity, include a figure in section 2 showing the lift-off of the degeneracy on the horizontally stretched Si layer.
3 – Please, explain better the concept on “antenna”. I do not see that a true antenna was simulated. Were the input impedance of the antenna A and the output impedance in A’ coupled to the simulation of the device?
4 – Please, explain the boundary conditions used and, particularly, the ones introduced in the “antennas”.
5 – Please, build a table to give all the parameters that were used in the simulations (for instance, the energy relaxation times, mobility models, etc.). Also introduce in this table the dimensions and doping levers of the different regions of the device.
6 – Is 0.25eV (line 154) a realistic value of the valleys split-off energy? What should the mole fraction of the Si1-xGex necessary to obtain such a level of split-off energy? What is the critical thickness of the strained Si to assume elastic strain (no deformation of the Si channel).
7 – As far as I understand, points in Figure 2 correspond to the authors calculations. Please, add results (experimental, Monte Carlo simulations, …) from literature and compare to the ones on Figure 2.
8 – Results in Figure 3 are obtained from simulations. Is the model able to take into account the intrinsic capacitances in the structure? Please, extract and plot Cgs and Cgd.
Along with the intrinsic capacitances, extrinsic ones do exist. Please, discuss if the capacitances (and inductances) may affect the phenomena and, at some extent, make the phenomenon non-observable.
English is essentially fine.
Author Response

(The authors gave the same response as above.)

Reviewer 3 Report
The manuscript (crystals-2596776) entitled " Numerical Simulations of Space Charge Waves Amplification Using Negative Differential Conductance in Strained Si/SiGe at 4.2 K " by Abel Garcia-Barrientos reported the a 2D numerical simulation of the amplification of space charge
waves using negative differential conductance in a strained Si/SiGe heterostructure at 4.2 K.
The manuscript is interesting and is well written.
I would recommend acceptance with minor revision.
(1) In Figure 1, “SiO2”, the oxygen atom is not showed correctly.
(2) All equations should be numbered.
(3) Figure 4, a and b are not well aligned.
Minor editing of English language required.
Author Response

(The authors gave the same response as above.)

Round 2
Reviewer 1 Report
No more comments
Reviewer 2 Report
The authors have satisfactorily answered the questions in my previous report. The revision by the authors has significantly improved the manuscript.